# Multitarget Reactive Magnetron Sputtering towards the Production of Strontium Molybdate Thin Films

**DOI:** 10.3390/ma16062175

**Published:** 2023-03-08

**Authors:** Mindaugas Andrulevičius, Evgenii Artiukh, Gunnar Suchaneck, Sitao Wang, Nikolai A. Sobolev, Gerald Gerlach, Asta Tamulevičienė, Brigita Abakevičienė, Sigitas Tamulevičius

**Affiliations:** 1Institute of Materials Science, Kaunas University of Technology, K. Baršausko St. 59, LT-51423 Kaunas, Lithuania; 2Solid State Electronics Laboratory, Technische Universität Dresden, 01062 Dresden, Germany; 3Departamento de Física and i3N, Universidade de Aveiro, 3810-193 Aveiro, Portugal

**Keywords:** strontium molybdate, multitarget reactive magnetron sputtering, X-ray photoelectron spectroscopy

## Abstract

X-ray photoelectron spectroscopy was used to study the direct synthesis of strontium and molybdenum oxide thin films deposited by multitarget reactive magnetron sputtering (MT-RMS). Sr and Mo targets with a purity of 99.9% and 99.5%, respectively, were co-sputtered in an argon–oxygen gas mixture. The chamber was provided with an oxygen background flow plus an additional controlled oxygen supply to each of the targets. We demonstrate that variation in the power applied to the Mo target during MT-RMS enables the production of strontium and molybdenum oxide films with variable concentrations of Mo atoms. Both molybdenum and strontium were found in the oxidized state, and no metallic peaks were detected. The deconvoluted high-resolution XPS spectra of molybdenum revealed the presence of several Mo 3d peaks, which indicates molybdenum bonds in a lower valence state. Contrary to the Mo spectra, the high-resolution strontium Sr 3d spectra for the same samples were very similar, and no additional peaks were detected.

## 1. Introduction

Strontium molybdate SrMoO_x_ (SMO) belongs to the SrO–Fe_3_O_4_–MoO_3_ system which exhibits a rich phase diagram [1]. Both SrMoO_3_ and SrMoO_4_ are compounds with a perovskite-type crystal structure (space group *I*4_1_/*a*) [2]. SrMoO_3_ and SrMoO_4_ have been shown to be transformed to each other via redox annealing [3]. SrMoO_3_ demonstrates rather good chemical and thermal stability [4] and is a promising candidate for plasmonic applications [3]. However, these require a high-quality material which is difficult to obtain due to unavoidable defects and nonstoichiometry peculiar to oxides. The defect content in a thin film depends on the deposition technique and respective growth parameters such as partial oxygen pressure, deposition rate, and target temperature, among others. The presence of the additional SrMoO_4_ crystalline phase in the SrMoO_3_ thin film acts to suppress the high-temperature degradation mechanism, namely the loss of crystallinity upon heating, that is present in single-phase SrMoO_3_ samples [5].

In the past, SMO thin films were deposited by chemical solution deposition on silicon substrates [6], via radiofrequency magnetron sputtering onto quartz substrates [7] and by pulsed laser deposition on substrates of strontium titanate (SrTiO_3_), magnesium oxide (MgO), and lanthanum aluminate (LaAlO_3_) [3,5]. As an alternative method for SMO film production, reactive magnetron sputtering could be used. To the best of our knowledge, we are not aware of any study on the direct synthesis of SMO via multitarget reactive magnetron sputtering (MT-RMS).

The MT-RMS of perovskite oxides was introduced in 1985 for the fabrication of electro-optic epitaxial (Pb,La)TiO_3_ thin films [8]. MT-RMS is a simple thin-film deposition technique that meets the requirements of device fabrication at an industrial scale. Since the late 1980s, it has been widely used for the manufacturing of ferroelectric PbTiO_3_- and Pb(Zr,Ti)O_3_-based thin films for applications such as piezoelectric sensors and actuators, ferroelectric memories, infrared devices, and other. The main advantages of multitarget sputtering are as follows [9]:Thin films with a precisely controlled composition can be obtained, and the stoichiometry of the films can be easily varied by changing the power supplied to the target; e.g., a possible loss of metal due to elevated temperatures can be compensated in this way;Stoichiometric variations on the target surface during repeated use can be prevented by target preconditioning and operation in the transition mode;Sputtering from metals in the transition region where the target racetrack is in the metallic mode and the metal is oxidized by the reactive gas on the way to the substrate provides a higher deposition rate;Compatibility with the current silicon technology.

It should be noted that point three requires precise control of the partial oxygen pressure. For this purpose, the plasma emission monitor (PEM) technique can be used, which is an already well-established process control tool for large-area reactive sputter deposition [10].

Especially, the Mo valence state is an indicator of the oxidation state and thus the oxygen stoichiometry and overall quality of the material. This valence state may be safely determined with X-ray photoelectron spectroscopy (XPS), which is widely used for surface chemical composition analysis and chemical bonds detection [11,12]. It is a surface-sensitive technique with an approximate measurement depth of about 1–10 nm. The sensitivity of this method depends on the element of interest and, generally, an atomic concentration above 1% is sufficient for analysis. Chemical bonds are analyzed using known binding energies for specific chemical compounds and detected peak positions in the spectra.

In this work, we apply XPS to study the valence state of Mo in mixed SrO-MoO_3_ oxide films in a wide range of Mo/Sr ratios. For the deposition of SrO-MoO_3_ oxide films with variable concentrations of Mo atoms, a variation in the power applied to the Mo target during MT-RMS was used. The obtained SrO-MoO_3_ oxide films can be used as a starting material for the production of SMO films by means of thermal treatment. In addition, techniques of Energy Dispersive X-Ray Spectroscopy (EDX), Scanning Electron Microscopy (SEM), Raman Scattering Spectroscopy, and X-ray Diffraction (XRD) were applied to obtain additional information on the composition and structure of the deposited samples.

## 2. Materials and Methods

### 2.1. Film Deposition

MT-RMS deposition was performed in an industrial sputtering system (LS730S, VON ARDENNE, Dresden, Germany). The computer-controlled system was equipped with a process chamber and a load lock, which were both evacuated by turbomolecular pumps. The background pressure in the process chamber was below 1.2 × 10^–4^ Pa. Pressure control was performed through a butterfly valve, and the deposition process was controlled by a PEM. The process chamber consisted of four metallic target positions with 200 mm diameters that were arranged circularly and directed towards a rotating substrate table. The latter was designed to hold four 150 mm wafers placed face down in a circle.

For the SrO-MoO_3_ mixed-oxide thin-film deposition, Sr and Mo targets with a purity of 99.9 and 99.5%, respectively, were co-sputtered in an argon–oxygen gas mixture. The chamber was provided with an oxygen background flow. Additional oxygen gas channels, controlled by piezoelectric valves, were introduced near each target surface. The gas flow of one channel was controlled by a closed-loop feedback circuit of the PEM. For the XPS analysis, Rubalit 701 alumina ceramic substrates (CeramTec, Marktredwitz, Germany) were mounted onto silicon wafers. To prevent arcing during reactive sputtering, an arc suppression (Advanced Energy Pinnacle, Advanced Energy, Denver, CO, USA) was connected between the power supplies and the Mo target. Sample numbers and deposition process parameters are presented in Table 1. All samples were fabricated by connecting the Mo target to a unipolar/bipolar switching unit UBS-C2 (Fraunhofer FEP, Dresden, Germany) to provide current pulses of 20 A with a duty of ~50% at a repletion frequency of 30 kHz. The Sr target was connected to a radiofrequency (RF) (13.56 MHz) generator. An upper limit of the film thickness was estimated by means of the film deposition rate of 1.8 nm/min determined using the reflection interferometry of thin films deposited similarly to sample #140819 onto platinized silicon wafers.

### 2.2. X-ray Photoelectron Spectroscopy

For the surface analysis, a KRATOS ANALYTICAL XSAM800 spectrometer employing nonmonochromatized Al K_α_ radiation (*hν* = 1486.6 eV) was used. The base pressure in the analytical chamber was lower than 2 × 10^–7^ Pa. The energy scale of the system was calibrated according to the Au 4f_7/2_, Cu 2p_3/2_, and Ag 3d_5/2_ peak positions. The C 1s, O 1s, Mo 3d, and Sr 3d spectra were acquired at the 20 eV pass energy and 0.1 eV energy steps (for survey spectra, 40 eV pass energy and 0.5 eV energy steps were used) and with the analyzer working in the fixed analyzer transmission (FAT) mode. After Shirley’s background subtraction, relative atomic concentrations were calculated from the peak areas using the original KRATOS software and relative sensitivity factors. Spectra deconvolution was performed by employing the XPSPEAK 4.1v software, where a symmetrical peak shape and a product of Gaussian and Lorentzian functions with a 70 to 30 ratio were used.

### 2.3. EDX and Scanning Electron Microscopy

A composition and structural analysis was conducted via Energy Dispersive X-ray Spectroscopy (EDX) and Scanning Electron Microscopy and was carried out in a Zeiss Supra 40VP facility (Oberkochen, Germany). EDX was performed using an EDAX Apollo 10SDD detector (AMETEC Inc., Berwyn, PA, USA).

### 2.4. Raman Scattering Spectroscopy

Raman spectra were recorded with an inVia spectrometer (Renishaw, UK) equipped with an optical microscope (Leica, Wetzlar, Germany) and a motorized sample stage. The excitation beam from a diode laser emitting at 532 nm was focused on the sample using a 50× objective. The laser power at the sample surface was 3.2 mW. The Raman Stokes signal was dispersed with a diffraction grating (2400 grooves/mm), and the data were recorded using a Peltier-cooled charge-coupled device (CCD) detector (1024 × 256 pixels). This system yields a spectral resolution of about 1 cm^−1^. Silicon was employed to calibrate the Raman setup in both Raman wavenumber and spectral intensity.

### 2.5. X-ray Diffraction

The crystal structure of the thin films on alumina ceramic substrates was determined using a D8 Discover X-ray diffractometer (Bruker AXS GmbH, Billerica, MA, USA) with a Cu K_α_ (*λ* = 1.5418 Å) radiation source and parallel beam geometry with a 60 mm Göbel mirror. A Soller slit with an axial divergence of 2.5° was utilized on the primary side. Diffraction patterns were recorded using a fast-counting LynxEye (0D mode) Silicon Strip detector with a 2.475° opening angle and 6 mm slit opening. The peak intensities were scanned over the range of 10–80° (coupled *θ–2θ* scans) with a 0.02° step size.

## 3. Results and Discussion

### 3.1. Film Microstructure and Bulk Composition

Figure 1 illustrates the surface structure of the fine-grained thin films in the same area where the EDX analysis was performed. The bulk chemical composition determined by EDX is given in Table 2.

### 3.2. Survey XPS Spectra and Atomic Concentrations

For the analysis of the surface composition of the films, atomic concentrations were calculated for each sample using the XPS peak area of the element and the corresponding sensitivity factors provided by the instrument manufacturer.

The results presented in Table 3 show that the oxygen content was rather high for most of the samples and was not particularly related to the Mo concentration. The presence of carbon and some part of oxygen could be due to the usual atmospheric contaminants and backstreaming of the pump oil. The strontium concentration was rather low, about one per cent or lower, and was close to the detection limit, except for sample #250719. In the bottom row, the relative concentration of Mo^6+^ bonds calculated from the deconvoluted Mo 3d spectrum is presented and described in the text below.

The survey XPS spectrum in Figure 2 shows all the detected peaks of the respective elements present in the film. The Sr 3d peak had a very weak intensity in this spectrum due to the low Sr content. The survey spectra for all the samples were similar, except for the Sr 3d peak, which was more pronounced for the samples with a higher Sr concentration (Appendix A).

A comparison of the normalized intensities of the high-resolution Mo 3d XPS spectra before deconvolution for all samples (Appendix A) indicated the presence of lower valence states of molybdenum. To investigate the molybdenum valence state, the deconvolution of Mo 3d spectra was performed for all samples (Figure 3 and Appendix A).

### 3.3. High-Resolution Spectra Analysis

A comparison of the high-resolution spectra after deconvolution (Figure 3) for sample #140819 (18.8% Mo) and sample #310719 (12% Mo) showed a typical Mo 3d spin-orbit doublet. The indicated position of the Mo 3d_5/2_ peak at 232.8 eV coincided well with the known position (232.8 eV, 232.6 eV) of Mo^6+^ in the MoO_3_ trioxide [11,13]. Lower-intensity peaks at smaller binding energies represent oxidized molybdenum in lower valence states. We assigned the peak at approximately 229.6 eV to Mo^4+^ bonds since its binding energy corresponds to the known value of 229.6 eV found in molybdenum dioxide MoO_2_ [11]. The most probable cause of the content variation in Mo suboxides is due to the deposition parameters used for the film growth, namely the oxygen flow. The highest suboxide (MoO_2_) content was found in sample #310719 (12% Mo concentration) which was produced using the lowest oxygen flow.

In order to further elucidate the valence of molybdenum, a simple curve approximation procedure (employing MS Excel software) was used. Initially, a graph (Figure 4) was constructed to represent the relationship between the molybdenum valence state and Mo 3d_5/2_ peak energy values (red circles) [11,12]. Then, all the points in the graph were used to draw an approximation curve (Figure 4, dashed line). Afterwards, the acquired binding energy of the fitted Mo 3d_5/2_ peak (Appendix A) was used to define the valence state of molybdenum (Figure 4, the violet arrows and asterisk on the curve). The relative concentrations of Mo bonds for all the samples calculated from spectra deconvolution are presented in Appendix A.

In Figure 5, the deconvoluted Sr 3d spectra for the same samples as in Figure 3 are presented. Both spectra were fitted with two overlapping peaks of a typical Sr 3d spin-orbit doublet. All the Sr 3d XPS spectra for all the samples were similar (Figure 5 and Appendix A), and the main peak position coincided with the reported Sr 3d_5/2_ binding energy value (133.5 eV) for oxidized strontium [5,13]. No additional peaks were detected.

A comparison of the oxygen XPS spectra in the O 1s region for some samples is shown in Appendix A. The deconvolution of the spectra for these samples revealed a similar structure consisting mostly of a peak at approximately 530.5 eV, which represents oxygen bound to molybdenum (530.5 eV) or oxygen bound to strontium (530.4 eV) [11,12]. A small amount of typical atmospheric contaminants (C-O and H-O bonds) [11,12,14] was also detected.

### 3.4. Raman Scattering Spectroscopy Analysis

The analysis of Raman spectra revealed that sample #150819 exhibited pronounced peaks corresponding to the MoO_3_ phase [15,16], as indicated by the squares in Figure 6 and Appendix A. The most intense bands at 994 cm^−1^ and 818 cm^−1^ corresponded to Mo-O stretching modes [16], whereas other peaks marked with squares in the spectra can also be attributed to different Mo-O vibrations as indicated in Ref. [17].

The spectra of samples #260719, #310719, #020819, and #120819 (not shown here) were too noisy to distinguish visible peaks corresponding to molybdenum oxide. It could be related to the small Raman scattering cross-section of intermediate forms of molybdenum oxide [17].

### 3.5. X-ray Diffraction Analysis

The XRD patterns of sample #150819 and the Al_2_O_3_ substrate are shown in Figure 7. The XRD patterns indicated the presence of the MoO_3_ phase, as marked by the squares in Figure 6. The two peaks exhibited (020) and (040) Bragg reflections, and this was attributed to the orthorhombic structure of α-MoO_3_ according to the powder diffraction data (JCPDS card no. 05-0508). As for the rest of the samples (see the Appendix A), they were found mostly in an amorphous state.

## 4. Conclusions

(1)Survey XPS spectra showed all elements present in the deposited films and a small amount of atmospheric contaminants; in addition, the same elements were detected in EDX, Raman spectra, and XRD as well.(2)Both Mo and Sr were found to be in the oxidized state, and no metallic peaks were detected. The deconvoluted high-resolution XPS spectra of molybdenum revealed the presence of several Mo 3d peaks, which indicated molybdenum bonds in a lower valence state. Contrary to the molybdenum spectra, the high-resolution strontium Sr 3d spectra for the same samples were very similar, and no additional peaks were detected.(3)The oxygen spectra also showed a similar structure consisting mostly of a peak at approximately 530.5 eV, which represents oxygen bound to molybdenum (530.5 eV) or oxygen bound to strontium (530.4 eV). A small amount of typical atmospheric contaminants (C-O and H-O bonds) was also detected.(4)Raman Scattering Spectroscopy revealed the presence of the MoO_3_ phase in one of the deposited films.(5)The XRD patterns showed the presence of a crystalline MoO_3_ phase for the same sample, while the other samples were found in an amorphous state.(6)The deposition of SrO-MoO_3_ oxide films with variable concentrations of Mo atoms was achieved using a variation in power applied to the Mo target during MT-RMS. The obtained SrO-MoO_3_ oxide films can be used as a starting material to produce SMO films via subsequent thermal treatment.

## Figures and Tables

**Figure 1 materials-16-02175-f001:**
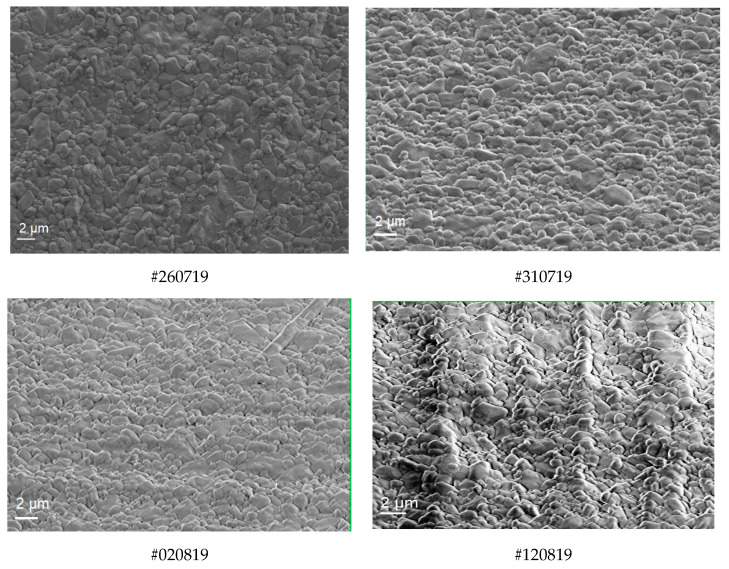
SEM images of samples #260719, #310719, #020819, and #120819.

**Figure 2 materials-16-02175-f002:**
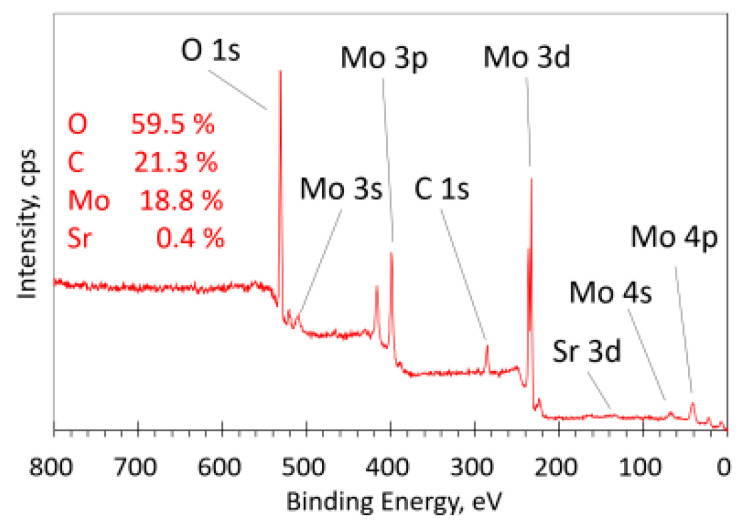
XPS survey spectrum of the film with 18.8% Mo for sample #140819.

**Figure 3 materials-16-02175-f003:**
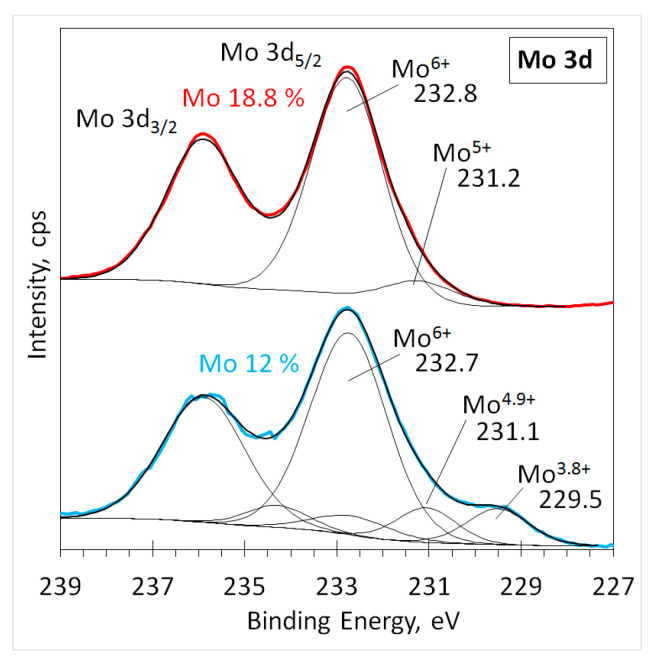
Deconvoluted high-resolution XPS spectra in the Mo 3d region for the samples with 18.8% and 12% Mo concentration. Thin black lines are fitted peaks, the thick black line is the envelope, and thick color lines are acquired spectra.

**Figure 4 materials-16-02175-f004:**
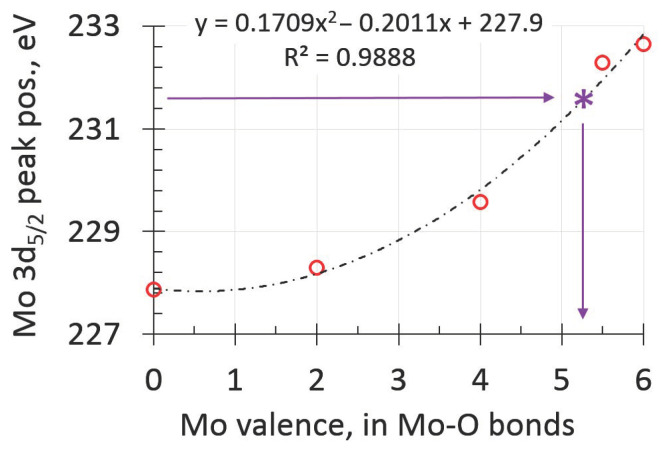
Relationship between the molybdenum valence state and Mo 3d_5/2_ peak energy values.

**Figure 5 materials-16-02175-f005:**
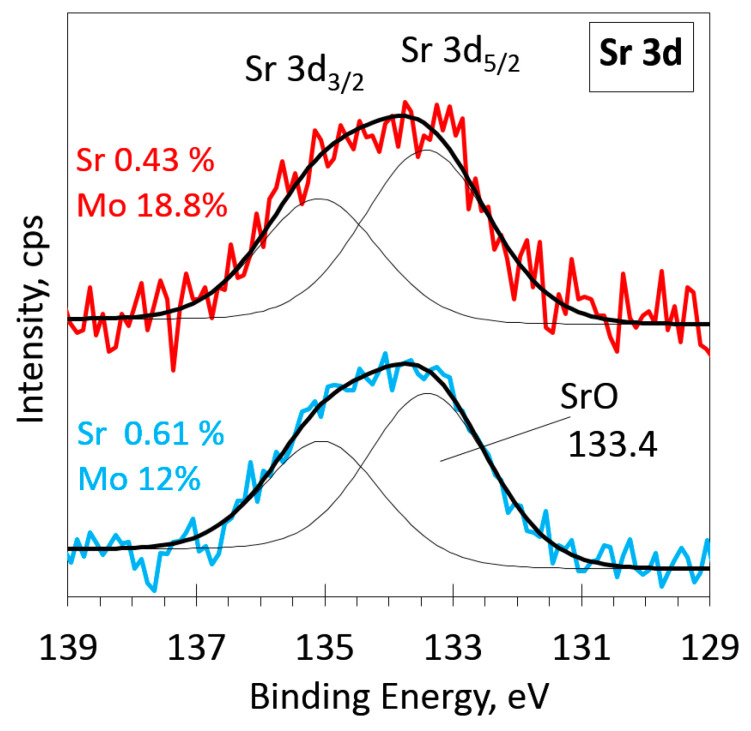
Deconvolution of high-resolution XPS spectra in the Sr 3d region. Thin black lines are fitted peaks, the thick black line is the envelope, and thick color lines are acquired spectra.

**Figure 6 materials-16-02175-f006:**
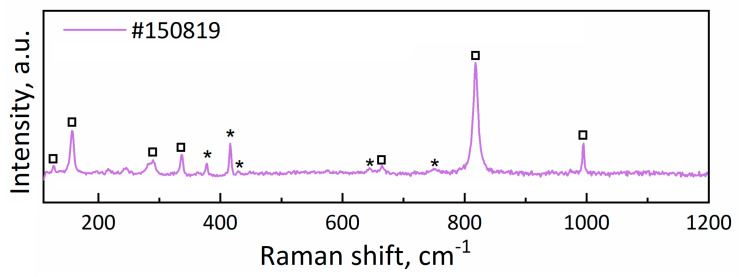
Raman scattering spectrum of sample #150819. The peaks corresponding to the MoO_3_ phase are indicated by squares, and those of Al_2_O_3_ are marked with asterisks.

**Figure 7 materials-16-02175-f007:**
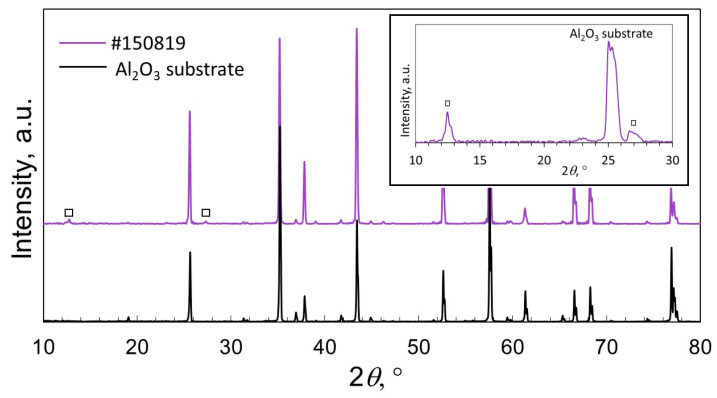
XRD patterns of sample #150819 and the Al_2_O_3_ substrate. The inset shows the XRD pattern collected for sample #150819 using a grazing incidence angle of 0.5 deg. The peaks corresponding to the MoO_3_ phase are indicated by squares.

**Table 1 materials-16-02175-t001:** Sample numbers and parameters of the deposition process.

Sample Number	#250719	#260719	#310719	#020819	#120819	#140819	#150819
Power at Sr target, W	300	300	300	300	320	320	320
O_2_ flow at Sr target, sccm	1	1	0	0.5	1.5	0.5	0.5
Power at Mo target, W	30	100	50	50	40	150	150
O_2_ flow at Mo target, sccm	1	1	0	0.5	1.5	0.5	0.5
Background O_2_ flow, sccm	5.5	4	0	1	5.5	0	5.5
Substrate temperature, °C	600	600	600	600	600	600	600
Deposition time, h	3	3	3	3	6	8	6 ^1^

^1^ not including a presputtering process.

**Table 2 materials-16-02175-t002:** Bulk atomic concentrations calculated using EDX data.

Element	#250719	#260719	#310719	#020819	#120819
O, %	99.31	98.53	96.00	99.70	99.74
Mo, %	0.17	1.37	3.77	0.19	0.17
Sr, %	0.26	0.09	0.16	0.11	0.09

**Table 3 materials-16-02175-t003:** Calculated surface atomic concentrations and estimated film thickness.

Element	#250719	#260719	#310719	#020819	#120819	#140819	#150819
O, %	45.51	59.75	51.95	76.63	80.36	59.45	64.85
C, %	45.07	24.34	35.44	18.59	14.42	21.28	17.93
Mo, %	4.29	14.79	12.00	3.8	4.39	18.84	15.91
Sr, %	5.13	1.12	0.61	0.97	0.82	0.43	1.32
Mo^6+^, %	94.7	89	78.6	77.2	75.7	89.9	94.35
D^1^, nm	324	324	324	324	648	846	648

D^1^—estimated film thickness, nm.

## Data Availability

The original contributions presented in this study are included in the article/Appendix A; further inquiries can be directed to the corresponding author.

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
