# Peer review of "Multitarget Reactive Magnetron Sputtering towards the Production of Strontium Molybdate Thin Films"

_materials, 2023, doi:10.3390/ma16062175_

Round 1

Reviewer 1 Report

This manuscript aims at the deposition of strontium molybdate SrMoOx (SMO) thin films using multi-target reactive magnetron sputtering. The authors present XPS analyzes for sample series with different deposition parameters.

At the same time, there are several important problems with this article:

1. Structural analysis of the thin layers obtained is missing. The phase composition is unknown. Please add XRD, Raman or other data supporting the phase composition of the samples. Is it SMO or a phase separated mixture of SrO and MoO3 oxides?

2. There are no data on the morphology of the thin films, neither SEM nor AFM nor TEM measurements are presented.

3. There is no data on the quality and physical properties of the films obtained, e.g., for the proposed plasmonic applications, luminescence, mechanical properties, etc.

4. The thickness of the deposited films is not shown.

5. Oxygen and carbon data (Table 2, Figure S6) are not reliable as there is no evidence that the surface of the sample was cleaned in XPS, for example with an ion gun. The conclusions about atmospheric impurities from these spectra are not relevant since the surface of the oxide film is usually covered with various -OH, -CO, and other groups under ambient conditions. By simply cleaning the surface and scrubbing thoroughly with an ion gun, you can determine the concentration of O and C in most samples.

6. Finally, the authors should summarize the quality of the received SMO films (if they are SMO!) as this is stated as the goal of the manuscript.

Author Response

We are very grateful to the Reviewers for their detailed analysis of the manuscript and valuable comments.

Reviewer 1 Review Report (Round 1)

Q1.1. Structural analysis of the thin layers obtained is missing. The phase composition is unknown. Please add XRD, Raman or other data supporting the phase composition of the samples. Is it SMO or a phase separated mixture of SrO and MoO3 oxides?

A1.1. The phase composition of Sr and Mo oxides was analyzed using Raman scattering spectroscopy and XRD. The Raman spectra and the relevant text have been added to the manuscript. XRD analysis and relevant text have been added to the manuscript and Supplementary materials. The composition of the films was evaluated using EDX analysis. Table 2 and the relevant text have been added to the manuscript.

Q1.2. There are no data on the morphology of the thin films, neither SEM nor AFM nor TEM measurements are presented.

A1.2. The surface structure of the films was evaluated by means of SEM. The SEM images of the samples and the relevant text have been added to the manuscript.

Q1.3. There is no data on the quality and physical properties of the films obtained, e.g., for the proposed plasmonic applications, luminescence, mechanical properties, etc.

A1.3. The quality and physical properties of the films are the subject of our next article; the samples will be annealed to form films with desired quality and physical properties.

Q1.4. The thickness of the deposited films is not shown.

A1.4. Here we only made a crude estimate based on the deposition rate and time. The exact thickness of the deposited films will be evaluated in our next article, where the samples will be annealed to form films with desired quality and physical properties.

Q1.5. Oxygen and carbon data (Table 2, Figure S6) are not reliable as there is no evidence that the surface of the sample was cleaned in XPS, for example with an ion gun. The conclusions about atmospheric impurities from these spectra are not relevant since the surface of the oxide film is usually covered with various -OH, -CO, and other groups under ambient conditions. By simply cleaning the surface and scrubbing thoroughly with an ion gun, you can determine the concentration of O and C in most samples.

A1.5. To avoid the artefact effects, namely the well-known (see T.L. Barr, J. Phys. Chem. 82 (1978) 1801) preferential sputtering and resulting sputter reduction of Mo6+, the samples were not cleaned using an Ar+ ion gun. As these SMO films will be annealed in an oxygen atmosphere at high temperatures in the next stage of our research, we expect that any atmospheric contaminants adsorbed on the film surface will be removed.

Q1.6. Finally, the authors should summarize the quality of the received SMO films (if they are SMO!) as this is stated as the goal of the manuscript.

A1.6. The phase composition of Sr and Mo oxides was analyzed using Raman scattering spectroscopy and XRD. The Raman spectra and the relevant text have been added to the manuscript. XRD analysis and relevant text have been added to the manuscript and Supplementary materials. Most likely the deposited films are mostly amorphous (before annealing process), only one sample showed XRD peaks corresponding to MoO3.

Reviewer 2 Report

XPS Analysis alone cannot be accepted for publication. Do more characterizational studies to support the results. 

 Application of Strontium Molybdate Thin Films has to stated clearly. 

Author Response

We are very grateful to the Reviewers for their detailed analysis of the manuscript and valuable comments.

Reviewer 2 Review Report (Round 1)

Q2.1. XPS Analysis alone cannot be accepted for publication. Do more characterizational studies to support the results.

A2.1. Analysis using Raman scattering spectroscopy and SEM/EDX/XRD was added to the manuscript.

Q2.1.1. Application of Strontium Molybdate Thin Films has to stated clearly.

A2.1.1. The goal of the formation of SMO films is to achieve a film with the desired quality and physical properties. It can be accomplished by the deposition of initial SMO films using the multitarget magnetron deposition technique followed by an appropriate annealing process. At this stage of our research, we show that we are able to successfully deposit SMO films with various compositions. At the next stage, the samples will be annealed to form films with desired quality and physical properties, and this will be discussed in our next article.

Reviewer 3 Report

1. Introduction. “It is a surface-sensitive technique with an approximate measurement depth of about 5 nm.” This affirmation is not correct, because XPS analysis depth depends on KE of photoelectrons and is variable in the range of about 1 – 10 nm. If you report the value of 5 nm for some material, it is necessary to present the calculation of IMFP for corresponding photoelectrons or corresponding citation of other authors.

2. The graph in Fig. S3 is not very accurate, because there are more data in the literature on different Mo oxidation states. For example, see T.L. Barr, J. Phys. Chem. 82 (1978) 1801, where has been reported BE = 231.6 eV for Mo2O5. If you add this value, the final part of the curve will change. Therefore, your approximation could be valid, but it is not so accurate to indicate the oxidation state with 3 digits, such as 4.87 or 3.83, whereas the reliable values must be 4.9 and 3.8. Anyway, the exact determination of Mo suboxides is not so important, because it is much more interesting the variation of their content in different samples. How can you explain the maximum content of Mo(4+) in the sample with 12% of Mo reported in Fig.2 and Figs. S2, S4? Why the content of suboxides depends on the total content of Mo?

3. The Fig. S3 must be moved to the main manuscript, because it is important for the chemical state determination.

4. It would be useful to investigate by SEM and/or AFM the surface morphology of different samples, which could be related to the content of Mo suboxides. Without any surface images this paper seems quite poor, because only XPS results don’t provide the full information about the films of mixed oxides. Also the information on crystalline structure (i.e., XRD measurements) could be helpful in order to understand the variation of Mo chemical state.

5.  Conclusions. The first one is banal and not informative, therefore it must be eliminated. The last one is exaggerated with BE values of O1s, because it is impossible to determine them with precision of 0.01 eV. As the paper quality is poor, the same is valid also for the conclusions, because the explanation on the variation of Mo chemical state is not reported.

Author Response

We are very grateful to the Reviewers for their detailed analysis of the manuscript and valuable comments.

Reviewer 3 Review Report (Round 1)

Q3.1. Introduction. “It is a surface-sensitive technique with an approximate measurement depth of about 5 nm.” This affirmation is not correct, because XPS analysis depth depends on KE of photoelectrons and is variable in the range of about 1 – 10 nm. If you report the value of 5 nm for some material, it is necessary to present the calculation of IMFP for corresponding photoelectrons or corresponding citation of other authors.

A3.1. The text in the manuscript ‘... depth of approximately 5 nm’ was replaced by the text “…depth of about 1-10 nm”.

Q3.2. The graph in Fig. S3 is not very accurate, because there are more data in the literature on different Mo oxidation states. For example, see T.L. Barr, J. Phys. Chem. 82 (1978) 1801, where BE = 231.6 eV for Mo2O5. If you add this value, the final part of the curve will change. Therefore, your approximation could be valid, but it is not so accurate to indicate the oxidation state with 3 digits, such as 4.87 or 3.83, whereas the reliable values must be 4.9 and 3.8. Anyway, the exact determination of Mo suboxides is not so important, because it is much more interesting the variation of their content in different samples. How can you explain the maximum content of Mo(4+) in the sample with 12% of Mo reported in Fig.2 and Figs. S2, S4? Why the content of suboxides depends on the total content of Mo?

A3.2. The precision was reduced to the recommended two digits. The behavior of the content of suboxides in the film is still under investigation and will be clarified at the next stage of our research, namely after annealing the films at high temperatures.

Q3.3. The Fig. S3 must be moved to the main manuscript, because it is important for the chemical state determination.

A3.3.  Fig. S3 has been moved to the main manuscript.

Q3.4. It would be useful to investigate by SEM and/or AFM the surface morphology of different samples, which could be related to the content of Mo suboxides. Without any surface images this paper seems quite poor, because only XPS results don’t provide the full information about the films of mixed oxides. Also the information on crystalline structure (i.e., XRD measurements) could be helpful in order to understand the variation of Mo chemical state.

A3.4. The phase of Sr and Mo oxides was analyzed by applying Raman scattering spectroscopy. The Raman spectra and the relevant text have been added to the manuscript. The composition of the films was evaluated using EDX analysis. Table 2 and a related text have been added to the manuscript. The surface structure of the films was evaluated by means of SEM analysis. The SEM images of the samples and the relevant text have been added to the manuscript.

Q3.5. Conclusions. The first one is banal and not informative, therefore it must be eliminated. The last one is exaggerated with BE values of O1s, because it is impossible to determine them with precision of 0.01 eV. As the paper quality is poor, the same is valid also for the conclusions, because the explanation on the variation of Mo chemical state is not reported.

A3.5. The first conclusion was rewritten, adding the confirmation of the XP survey spectra analysis by the EDX/Raman/XRD results. In the third conclusion, the precision of binding energy values for O 1s spectra was reduced.

Round 2

Reviewer 1 Report

Authors have updated manuscript and supporting information revealing important details. 

Major conclusion is that the obtained thin films are a mixture of MnO3 and SrO, there is no sign of SrMoO4. Even more from the presented results authors conclude that there is a low concentration of Sr. In SMO Sr:Mn proportion is 1:1, so, there not a chance that film contains SMO. No other data indicate SMO presence there. 

This requires changes in title, abstract, goals of the research and  conclusions reflecting the real outcome of this research. Also there is a question about the practical and academic value of the obtained mixture of Mn and Sr oxides. 

Also in Figure 5 and Figure S4 there is a label SrMnO4 which is not correct, it should be SrO. 

Unfortunately after receiving answers I cannot recommend to publish this manuscript under present title “XPS Analysis of Strontium Molybdate Thin Films Obtained by Multi-target Reactive Magnetron Sputtering”, as strontium molibdate is not there.

Otherwise, if authors can present a value of Mn and Sr oxide mixture this is a valid research to publish. 

Author Response

Q1.1. Major conclusion is that the obtained thin films are a mixture of MnO3 and SrO, there is no sign of SrMoO4. Even more from the presented results authors conclude that there is a low concentration of Sr. In SMO Sr:Mn proportion is 1:1, so, there not a chance that film contains SMO. No other data indicate SMO presence there. This requires changes in title, abstract, goals of the research and conclusions reflecting the real outcome of this research. Also there is a question about the practical and academic value of the obtained mixture of Mn and Sr oxides. 

A1.1. The appropriate corrections to the manuscript were made: the title, abstract, goals and conclusions were modified. The practical and academic value of the obtained mixture of Mo and Sr oxides were explained. 

Q1.2. Also in Figure 5 and Figure S4 there is a label SrMnO4 which is not correct, it should be SrO. 

A1.2. The appropriate corrections to the manuscript were made.

Q1.3. Unfortunately after receiving answers I cannot recommend to publish this manuscript under present title “XPS Analysis of Strontium Molybdate Thin Films Obtained by Multi-target Reactive Magnetron Sputtering”, as strontium molibdate is not there.

A1.4. The appropriate corrections to the title of the manuscript were made:

Q1.4. Otherwise, if authors can present a value of Mn and Sr oxide mixture this is a valid research to publish. 

A1.4. The value of the obtained mixture of Mo and Sr oxides were added to the manuscript. 

Reviewer 2 Report

Accept

Author Response

We are deeply grateful to the Reviewer for critically evaluating of the manuscript.

Reviewer 3 Report

1. XRD results are very strange: only in one sample are observed the peaks of MoO3, whereas in all other samples only the peaks from substrate are registered. I suppose there was something wrong with experiment geometry, because the samples are thick enough (300 – 900 nm) for reliable XRD measurements. As these results are completely useless, they should be eliminated. The same is valid for EDX results in Table 2, where practically only oxygen is observed. Again, it is strange why the content of Mo is so low? Practically only oxygen in Al2O3 substrate is detected, even if the films are thick enough for EDX analysis.

2. Again, I can only repeat that the most important result of this manuscript is the content variation of Mo suboxides, but the explanation of this effect is completely absent. Is it related to different surface morphology shown in Figure 1 and different deposition parameters listed in Table 1?  

As the scientific quality of this manuscript is poor, I recommend to reject it and encourage the authors to resubmit after serious improvement of analytical experiments and discussion of obtained results.

Author Response

Q3.1. XRD results are very strange: only in one sample are observed the peaks of MoO3, whereas in all other samples only the peaks from substrate are registered. I suppose there was something wrong with experiment geometry, because the samples are thick enough (300 – 900 nm) for reliable XRD measurements. As these results are completely useless, they should be eliminated. The same is valid for EDX results in Table 2, where practically only oxygen is observed. Again, it is strange why the content of Mo is so low? Practically only oxygen in Al2O3 substrate is detected, even if the films are thick enough for EDX analysis.

A3.1. The XRD patterns showed the presence of crystalline MoO3 phase for the same sample, while rest of the samples were found in amorphous state. The respective text was added to the manuscript.

Q3.2. Again, I can only repeat that the most important result of this manuscript is the content variation of Mo suboxides, but the explanation of this effect is completely absent. Is it related to different surface morphology shown in Figure 1 and different deposition parameters listed in Table 1?  

A3.2. The most probable cause of the content variation of Mo suboxides is due to the deposition parameters used for film deposition, namely oxygen flow. The highest suboxide (MoO2) content was found in the sample #310719 (12% Mo concentration), produced using the lowest oxygen flow. The respective text was added to the manuscript.

Q3.3. As the scientific quality of this manuscript is poor, I recommend to reject it and encourage the authors to resubmit after serious improvement of analytical experiments and discussion of obtained results.

A3.3. The value of the obtained mixture of Mo and Sr oxides were added to the manuscript.